# Silent Reactivation of Varicella Zoster Virus in Pregnancy: Implications for Maintenance of Immunity to Varicella

**DOI:** 10.3390/v14071438

**Published:** 2022-06-30

**Authors:** Mirella Mourad, Michael Gershon, Satish K. Mehta, Brian E. Crucian, Nicole Hubbard, Jing Zhang, Anne Gershon

**Affiliations:** 1Department of Obstetrics and Gynecology, Columbia University Irving Medical Center, New York, NY 10032, USA; mjm2246@cumc.columbia.edu; 2Department of Pathology, Columbia University Vagelos College of P&S, New York, NY 10032, USA; mdg4@cumc.columbia.edu; 3JES Tech, Human Health and Performance Directorate, Houston, TX 77058, USA; satish.k.mehta@nasa.gov; 4National Aeronautics and Space Administration (NASA) Johnson Space Center, Human Health and Performance Directorate, Houston, TX 77058, USA; brian.crucian-1@nasa.gov; 5Infectious Disease Pathology, ProMedica Laboratory, Department of Pathology, University of Toledo College of Medicine and Life Sciences, Toledo, OH 43614, USA; nicole.hubbardmd@promedica.org; 6Department of Pathology, Columbia University Irving Medical Center, New York, NY 10032, USA; jz14@cumc.columbia.edu; 7Department of Pediatrics, Columbia University Irving Medical Center, New York, NY 10032, USA

**Keywords:** VZV, reactivation, latency, saliva, salivary VZV DNA

## Abstract

We encountered two cases of varicella occurring in newborn infants. Because the time between birth and the onset of the illness was much shorter than the varicella incubation period, the cases suggested that the infection was maternally acquired, despite the fact that neither mother experienced clinical zoster. Thus, we tested the hypothesis that VZV frequently reactivates asymptomatically in late pregnancy. The appearance of DNA-encoding VZV genes in saliva was used as an indicator of reactivation. Saliva was collected from 5 women in the first and 14 women in the third trimesters of pregnancy and analyzed at two different sites, at one using nested PCR and at the other using quantitative PCR (qPCR). No VZV DNA was detected at either site in the saliva of women during the first trimester; however, VZV DNA was detected in the majority of samples of saliva (11/12 examined by nested PCR; 7/10 examined by qPCR) during the third trimester. These observations suggest that VZV reactivation occurs commonly during the third trimester of pregnancy. It is possible that this phenomenon, which remains in most patients below the clinical threshold, provides an endogenous boost to immunity and, thus, is beneficial.

## 1. Introduction

Extreme stress, such as space flight, may cause varicella zoster virus (VZV) to reactivate from latency (in persons who have had varicella or have been vaccinated against it) without producing symptoms. This phenomenon was reported to occur in 50–65% of astronauts during and after short- and long-duration spaceflights [1,2] and in 17% of children who were immune to VZV and stressed by hospitalization in an intensive care unit (ICU) [3]. These reports employed the transient presence of VZV DNA in the saliva as an indication of viral reactivation. In part because the screening of saliva for the presence of VZV DNA is available only on a research basis, it is not clear what the incidence and prevalence are of asymptomatic reactivations of VZV in individuals who are immune to varicella. Although the frequency with which asymptomatic VZV reactivation occurs is unknown, published data imply that VZV DNA is rarely, if ever, detected in the saliva of healthy, young individuals [4].

We encountered two newborn infants who developed varicella just after birth, a timeframe that is much shorter than the incubation period for varicella. These observations suggested that VZV was passed from the mother to the fetus prior to delivery. Because neither mother displayed varicella or zoster, it was likely that asymptomatic reactivations of VZV occurred in late pregnancy and provided the virus that completed the varicella incubation period perinatally. Because the varicella in the newborn infants was mild, maternally derived immunity was also transferred and ameliorated the infection. These observations led us to ask whether the asymptomatic reactivation of VZV occurs frequently in pregnancy. To answer this question, we determined whether VZV DNA could be detected in the saliva of a group of pregnant women in the absence of symptoms, as has been reported in astronauts [1,2]. We also determined whether other herpesviruses might reactivate in pregnancy.

*Case 1*. This infant was born at Columbia Presbyterian Hospital in New York City in 2018. His mother, a 37-year-old healthy woman presented in active labor at 36 weeks of pregnancy. Her antenatal period was significant only for vaginal candidiasis. She had no history of febrile illnesses or rashes in the antenatal period and no chronic medical conditions. She recalled experiencing chickenpox in childhood and had not experienced clinical zoster. Her serum varicella IgG level was positive prior to pregnancy. Her male baby was delivered by Caesarean section for nonreassuring fetal heart rate with a rupture of membranes at delivery. The APGAR scores were 8 and 9; the infant’s weight was 3295 g. The infant was hypoglycemic and hypothermic shortly after delivery. Ampicillin and gentamicin were administered after blood cultures were obtained. The postnatal examination revealed a mild, diffuse, pustular rash with scattered petechiae. Dermatology was consulted; the differential diagnosis included systemic bacterial or viral infection and transient neonatal pustulosis. On day one of life, bacterial and fungal blood and skin cultures were obtained and were found to be negative. VZV and herpes simplex virus (HSV) cultures from skin lesions were obtained and were found to be negative. Acyclovir 60 mg/kg/dose IV was administered since the infant was thought possibly to have a neonatal HSV infection. Direct fluorescent antigen (DFA) tests from the skin lesions then tested positive for VZV and negative for HSV. The polymerase chain reaction (PCR) tests obtained from skin lesions were positive for VZV and negative for HSV. A diagnosis of varicella was made in the infant, although there was no obvious exposure of the baby to VZV. A lumbar puncture was not performed. The diagnosis of varicella was made in the infant, and treatment with acyclovir was continued for a week. The infant’s rash improved, and the baby appeared entirely well on a follow-up visit.

*Case 2*. This 39-week, 1-day-old male infant was born via Cesarian section due to failure to progress at Pro Medica Russell J. Ebeid Children’s Hospital in Toledo, Ohio, in June 2021. The pregnancy was complicated by gestational hypertension. He presented with vesicular lesions on his right cheek in the first 24 h after birth. Subsequently, a mild rash was noted on his trunk that was thought to be erythema toxicum. Neonatal HSV infection was suspected, but a PCR test on a lesion on the cheek was negative for HSV DNA. It was positive, however, for VZV DNA (tested at ProMedica Laboratories, as well as in the laboratory at Columbia). As part of routine prenatal testing, his mother was found to have a positive IgG immunoassay for VZV IgG in her first trimester of pregnancy. She recalled experiencing varicella as a child. Except for the rash, the baby appeared well. New lesions developed on day four of life. By day six, all the lesions had crusted, and no further lesions developed. VZV IgG testing was positive on the infant. The mother was asymptomatic throughout the illness of her infant but reported that an aunt who babysat her daughter experienced shingles prior to the delivery. The infant was discharged and went home at 11 days of age.

## 2. Methods

We performed a prospective observational study in order to determine if we could identify evidence of the asymptomatic reactivation of VZV in pregnancy. Pregnant subjects from different trimesters in pregnancy who received prenatal care at our institution were approached and consented to participate in our study. The subjects were recruited from 3 general areas: obstetric ultrasound, the antepartum unit, and the labor and delivery unit. The subjects were excluded if they had any signs or symptoms of COVID-19 infection or other viral illness. Saliva was collected with commercial kits for this purpose obtained from Omnigene-oral Genotek (Ottowa, ONT, Canada), as described previously [5]. Nested PCR was employed at Columbia University for the detection of VZV DNA, as previously described. Real-time PCR was performed at NASA [1], as well as at Columbia University. This study was approved by the Institutional Review Board at Columbia University Irving Medical Center (AAAS6203), Institution Review Board at Johnson Space Center, NASA; Study ID: Pro2773, and the University of Toledo College of Medicine and Life Sciences.

## 3. Results

Saliva samples were obtained from 19 pregnant women. These women ranged in age from 25 to 46 years (mean 35). Their ethnic backgrounds were varied and included White (n = 9), Black (n = 4), Hispanic (n = 5), and Asian (n = 1). Ten patients provided a history of having experienced varicella in childhood; three did not, and six could not recall. Three remembered having received the live attenuated varicella vaccine and did not have a history of varicella. Most patients were healthy at baseline; however, two patients had a chronic medical condition: one had cystic fibrosis, and one had systemic lupus erythematosus. Samples were obtained from five women in the first trimester of pregnancy (weeks 9–12), and fourteen from women in the third trimester (weeks 25–40; average 34 weeks).

The saliva from the patients was independently assayed in two different laboratories: at Columbia and at NASA. From the Columbia laboratory, 11/12 (92%) third-trimester patients were found to have VZV DNA in their saliva by nested PCR. In the NASA laboratory, 7/10 (70%) were positive by real-time PCR. Their viral loads ranged from 508 to 1128 copies/mL with a mean + SE of 717 + 84 copies/mL saliva. Ten saliva samples were tested at both sites; the results agreed in 8/10 samples (80%). Two third-trimester patients had positive saliva samples at Columbia but negative tests at NASA. Saliva from 5/5 women in the first trimester were negative by qPCR at Columbia; 2/2 tested were negative at NASA. At NASA, no reactivation of HSV or cytomegalovirus (CMV) was identified in saliva from nine patients by qPCR.

In summary, at Columbia, of 12 women who were studied in weeks 25–40, 11/12 had positive saliva for VZV DNA (92%) using nested PCR. At NASA, of 10 patients studied in weeks 25–40, 7/10 (70%) were positive for VZV DNA by qPCR. In five women whose saliva was studied during the first trimester for VZV DNA, 0/5 were positive (Appendix A).

## 4. Discussion

We became interested in whether the reactivation of VZV in pregnancy was common or not after observing two newborn infants with varicella. One reported infant was born at Columbia. The other was encountered 3 years later because the medical team responsible for the care of this infant had heard about the original Columbia patient and brought their patient to the attention of the Columbia physicians. The silent reactivation of VZV is of particular interest because it is a potential stimulant of maternal immunity to VZV. It is interesting that VZV was reactivated in the setting of decreased immunity in pregnancy, which is thought to occur so that the mother does not reject her fetus. Immunity to VZV, however, was not so diminished in the pregnant women that clinical zoster was able to occur. Reactivation was only identified in women during late pregnancy.

It is not fully understood why immunity to varicella persists for years, both in individuals who have experienced clinical varicella and also in those who have been vaccinated. Two theories exist [6,7]: One is that VZV may periodically and spontaneously reactivate from latency (with or without symptoms), leading to what is termed as internal boosting. The other is that periodic exposure to VZV in persons with varicella is required to maintain immunity, which is called exogenous boosting [8]. The demonstration of the asymptomatic reactivation of VZV from latency, therefore, has importance for vaccination programs, as well as being an interesting phenomenon in which newborn infants can, on rare occasions, develop mild varicella from their mothers.

The diagnosis of neonatal varicella in the first infant in this report led us to begin to determine whether the reactivation of VZV could be identified in late pregnancy, when an incipient mother presumably becomes at least somewhat immunocompromised. The second infant was born 3 years later, after the study was underway, at a different hospital. Both cases were demonstrated to be due to VZV by the PCR of skin specimens at Columbia. There is no question about the diagnosis of VZV in these infants. It was, at first, suspected that each baby might have a herpes simplex virus (HSV) infection, but HSV was not detected in either one by PCR, ruling out this possibility. The babies were treated appropriately with acyclovir and recovered. The course of their varicella was mild, as would also be expected, due to the presence in the infants of maternal antibodies to VZV (but not enough to prevent the clinical infection of the infants) and antiviral therapy.

When these young babies were diagnosed with varicella, it was clear that the VZV had to have come from their mothers, who themselves had no symptoms of VZV infection. The shortest possible incubation period of varicella is 7 days, so postnatal exposure was not possible. The phenomenon of varicella in 1- to 3-day old infants was described previously in three newborns many years ago [9]. In all the instances, the illness in the baby was mild. It was concluded that the mother had to be the source of the virus, but the mechanism by which this might have transpired was neither clear nor was it even discussed in older publications. We now postulate that the reactivation of VZV occurred in these mothers, who must have experienced an asymptomatic form of herpes zoster similar to that described in astronauts [1,2].

In the current study, we found that the silent reactivation of VZV during pregnancy was common. This was shown independently in two different laboratories. There was close agreement between the two laboratories for women in the third trimester (92% reactivation at Columbia vs. 70% at NASA). Nested PCR is more sensitive than qPCR, so this result was not unexpected. It is of interest that the saliva samples of five women tested in early pregnancy were negative for reactivation by qPCR at both laboratories.

In addition to the rare cases of clinical varicella in neonates who were born to healthy mothers that we described, other data suggest that VZV from healthy mothers can cross the placenta. About 20 years ago, VZV was reported in the cerebrospinal fluid (CSF) of two neonates with neurologic abnormalities and no history of rash. These infants were born to healthy mothers who completed normal pregnancies [10,11]. It was reasoned that the virus that infected the children might have been derived from a reactivation of maternal VZV during pregnancy leading to viremia, which was asymptomatic in the mothers but, nevertheless, crossed the placenta to enter the infants who, as a result, developed serious neurological abnormalities. More recently, the transfer of VZV DNA from 61 pregnant mothers to approximately one-third of their infants was reported in a study of sera obtained from pregnant women and their infants [12]. In these cases, however, there were no associated symptoms in the infants.

We found no evidence in our study on saliva of reactivation of HSV or cytomegalovirus (CMV) in the pregnant women we described. We expected that we might find reactivation of other herpesviruses, such CMV and HSV, which reactivate along with VZV in astronauts [2], but we did not.

About 50 years ago, Hope-Simpson proposed the hypothesis that VZV could reactivate from latency to cause zoster [6]. In addition, Hope-Simpson also proposed that VZV would periodically reactivate silently, liberating an amount of the virus that was too low to cause clinical disease but would, nevertheless, lead to episodes of asymptomatic viremia. His idea was that these asymptomatic reactivations would boost an individual’s immunity to the virus. Despite this mechanism, however, immunity could still wane as a function of age, leading to the manifestation of clinical zoster [6]. Implicit in Hope-Simpson’s hypothesis was that the silent reactivation of VZV could stimulate immunity to the virus. At the time of Hope-Simpson’s publication, humoral immune responses were recognized, but cellular immunity to VZV had not yet been described. At that time, moreover, Hope-Simpson was unable to provide data to verify that his postulated asymptomatic reactivation of VZV actually did occur. Usually, in 70% of people over a lifetime, cellular immunity prevents VZV from reactivating and causing clinical zoster [13]. When cell-mediated immune function declines, however, as it does in the elderly, VZV may reactivate in neurons that project to the skin, causing symptomatic zoster with a classical skin rash [14]. This occurrence is well-known, but it is not evidence of the asymptomatic reactivation of VZV.

The periodic silent reactivation of VZV without symptoms that Hope-Simpson postulated could be a critical factor in the maintenance of long-term immunity to VZV. After primary infection, VZV becomes latent in peripheral neurons, including those of the enteric nervous system (ENS) [5,7]; moreover, reactivation in enteric neurons would be a strategic location to stimulate immunity because the gastrointestinal tract is the largest organ of the immune system in the body [15]. Since enteric neurons do not project to the skin, the reactivation of VZV in the gut can occur in the absence of skin lesions and, often, without provoking symptoms [5,16]. It is, thus, important to recognize and understand silent VZV reactivation because it may play a role in maintaining long-term immunity to VZV. Not only does wild-type VZV become latent after primary infection, but so also does the vOka strain of the live attenuated varicella vaccine [7,17].

Some epidemiological investigators assumed that exogenous exposure to circulating wild-type VZV is critical for the maintenance of immunity to VZV and built computer models to predict what would happen if VZV ceased to circulate and epidemics of varicella no longer occurred [8]. These models predicted that epidemics of zoster and a loss of immunity to varicella would be the inevitable long-term outcomes of decreased exposure to circulating wild-type VZV. The assumptions, however, that formed the basis of the dire predictions of the computer models did not include the possible effects of silent reactivation of VZV with the stimulation of immunity by endogenous exposure to VZV (endogenous boosting). Fortunately, enough time has passed since the licensure of the live attenuated VZV vaccine in the US, its nearly universal use, and the virtual disappearance of circulating VZV in the USA to ask whether the predicted epidemics of zoster have occurred. In fact, there is no convincing evidence that the incidence of zoster is greater in countries that routinely vaccinate against varicella than in those that do not [18]. Laboratory evidence of immunity to VZV in older adults is similar in countries that do or do not routinely use a varicella vaccine [19,20]. The incidence of zoster in cloistered nuns and monks who are not exposed to children and their epidemics of varicella, moreover, is not different from that in the general population [21]. The failure of the disastrous spread of zoster to occur suggests that endogenous boosting is important in maintaining immunity to VZV. It is, therefore, important to document instances of inapparent VZV reactivation. Such occurrences may be more frequent than is appreciated. Clearly, stress can reactivate VZV without symptoms, as has been reported in astronauts during and after space travel [1,2]. The reactivation of VZV without symptoms could be common due to the stresses of late pregnancy and delivery. Mothers are young and ought to have healthy, well-functioning immune systems. Such a group might be expected to have sufficient cellular immunity to prevent reactivated VZV from reaching the threshold necessary to manifest clinical zoster. In most cases, their immunity would also be sufficient to protect the fetus. The rare escape of reactivated VZV from immune surveillance might give rise to neonatal varicella, but in most instances, VZV reactivation during late pregnancy is probably beneficial to the mother and even to her child. The event would be silent and asymptomatic, but immunity to VZV, including protection against zoster, would receive a boost.

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
