# Peer review of "Silent Reactivation of Varicella Zoster Virus in Pregnancy: Implications for Maintenance of Immunity to Varicella"

_viruses, 2022, doi:10.3390/v14071438_

Round 1

Reviewer 1 Report

This brief article documents convincing evidence that pregnancy can trigger the subclinical reactivation of latent VZV, which on some occasions, can give rise to virtual immediate onset of VZV infection in newborns without prior "incubation" periods. I found the argument of considerable interest and predict this may open up a large new subject on the state of VZV in newborns following the stress of pregnancy and delivery.  there are few corrections comments

  1. I am please to see that the PCR results were confirmed by two different groups and by two different approaches.     Had the results relied on the nested PCR approach only,  i would have been far more sceptical of the data. Nested PCR is an open tube design mid way through that is easily highly susceptible to contamination issues.  The finding of DNA positivity by Q PCR is more convincing. Can this data be presented in a table so readers can see how the PCR of controls versus candidates varied? 
  2. the study was done on some patients who had been vaccinated with vOka.  It is quite controversial if vOka reactivates commonly in an asymptomatic manner.  But it was not clear if the virus was WT or vaccine by PCR, please comment.   This certainly will become an important future issue, and i would recommend the authors consider the varivax vaccinated  population who are now reaching the age at which many are going through pregnancy.  Agreed, its a future study, but would be an important aspect to consider. Maybe add to discussion after line 210.  Again, at line 105, were the vaccinated individuals positive for vaccine or for WT virus?  This does seems kind of important.      
  3. line 159 and 195 the term asymptomatic Herpes zoster is used. This is not a term i would use, as herpes zoster is usually largely defined by the clinical presentation of the skin disease and or the pain associated with it. Even zoster sine herpete is usually a result of pain signs.  I am fine with the term asymptomatic reactivation.  
  4. paper needs an abstract
  5. line 194, cellular imminity seems to prevent HZ, yes.  whether it prevents reactivation is part of the issue being discussed here.  Maybe reword to "reactivating to cause herpes zoster"
  6. line 54 define "Shortly"  if possible 

Author Response

We have made the changes suggested by the reviewer.  We have not determined if Oka VZV is present in saliva of pregnant women.  We will do this in the future.An excellent suggestion.

We have included an abstract.

We have included the other suggestions by the reviewer.

Reviewer 2 Report

In this manuscript entitled "Silent Reactivation of Varicella Zoster Virus in Pregnancy: Implications for Maintenance of Immunity to Varicella", Gershon and her colleagues advanced the concept of silent reactivation of VZV which has been initially proposed by identification of this phenomenon in astronauts. Inspired by two cases of mild varicella just after delivering in infants whose mothers did not have symptomatic VZV reactivation, they examined VZV DNA in saliva of 19 pregnant women and found that most women during the third trimester but not the first trimester shed the VZV (DNA) as an indicator of silent reactivation of VZV but not HSV or HCMV. Totally results seem valid and conclusion is supported by the results and also backed up by previous published findings, however, the manuscript can be improved by adding summary Table of results.

Comments are below.

  1. In case 1: there are many technical terms abbreviated or just shortened used during daily conversation during working especially diagnostic examinations or relating with obstetrics, gynecology and pediatrics. Please write full terminology as possible; e.g. "bacterial and fungal blood" might be better "bacterial and fungal culture tests from blood sample"? (not limited to these). Also what is APGER score and what this indicates (easy to search but nice if adding one sentence for wider virology community)?
  2. Line 51: "varicella IgG" should be "VZV IgG".
  3. Please generate summary Table for results.
  4. Line 164: The statement about superiority of nested PCR to qPCR is very subjective. There might be many factors to believe so but for the reviewer it seems just not optimize methodology. Otherwise this statement should encourage the authors to test nested PCR for five VZV negative samples during the first trimester which were negative by the nested PCR, while the reviewer does not think the nested PCR is essential.

Author Response

We have added a table showing our results, as requested.

Reviewer 3 Report

The main findings of the study relies on the investigation of Varicella Zoster Virus genomic sequences in saliva samples from two small cohorts of (n=5) women in the first and (n=14) women in the third trimester of pregnancy and analyzed at two different laboratories with different methods, i.e., nested PCR, at quantitative PCR. No VZV DNA was detected at either site in saliva or women during the first trimester; however, VZV DNA was detected in a majority of samples of saliva (11/12 examined by nested PCR; 7/10 examined by qPCR) during the third trimester. These observations suggest that VZV reactivation occurs commonly during the third trimester of pregnancy.

The main limitation of the study is the reduced sample size. 

Here several suggestions for improving the manuscript

1.    The abstract should be included in the main file. I can read it in the submission system, only. 
2.    The introduction should be reorganized for a better readability. The description of the two cases should be reduced. The aim of the study should also be included at the end of the introduction. 
3.    Lines 22-24 the sentence should be improved in readability
4.    Line 33 please remove the parenthesis 
5.    Lines 125-130, the physiological immune modulation during pregnancy may result in a more increased susceptibility to viral infections, as being theorized for Merkel cell polyomavirus DOI: 10.3389/fmicb.2021.789991. This more susceptibility may lead to an increase in viral replication levels. This information should be included
6.    Details on DNA isolation should be included in the methods, including tissue of origin (saliva samples)
7.    Lines 98-160 these lines should be moved to the methods section
8.    Line 11 “mean + SE” better mean + standard error (SE)
9.    Discussion, lines 130-139 details on Varicella-Zoster Virus reactivation are also reported here PMID: 29895192 and here PMID: 34034659
10.    Line 166 nested PCR seems to be prone to generate false positive results comaperd to quantitative PCR PMID: 16678378. Moreover, the more sensitive droplet digital PCR can also be considered as an alternative methodological approach for detecting VZV genomic sequences ttps://doi.org/10.1111/ceo.13440
11.    Line 169 better “ An early study reported the presence of vzv…”
12.    Study limitations should be included.
13.    Study conclusions should be included at the and of the discussion

Author Response

We have included a table of our results... PCR and NASA realtime PCR